# Comparison of Immunodiagnostic Assays for the Rapid Diagnosis of Coccidioidomycosis in Dogs

**DOI:** 10.3390/jof8070728

**Published:** 2022-07-13

**Authors:** Diego H. Caceres, Mark D. Lindsley

**Affiliations:** 1Mycotic Diseases Branch, Centers for Disease Control and Prevention (CDC), Atlanta, GA 30333, USA; 2Center of Expertise in Mycology Radboudumc/CWZ, 6525GA Nijmegen, The Netherlands; 3Studies in Translational Microbiology and Emerging Diseases (MICROS) Research Group, School of Medicine and Health Sciences, Universidad del Rosario, Bogotá 1653, Colombia

**Keywords:** coccidiodiomycosis, *Coccidioides*, dogs, antibody, immunodiagnosis, lateral flow assay, enzyme immunoassay, immunodiffusion

## Abstract

Coccidioidomycosis is a disease caused by the dimorphic fungi *Coccidioides* spp., which affects humans and a variety of animal species, including domestic dogs. In dogs, accurate diagnosis could provide a substantial improvement on the quality of canine life, as well as an advancement in the mapping of regions endemic for coccidioidomycosis. The purpose of this study was to compare immunodiagnostic assays for anti-*Coccidioides* antibody (Ab) detection in dogs’ serum. Three commercially available immunodiagnostic assays (IMMY^®^; Norman, OK, USA) were evaluated, including the sōna *Coccidioides* Ab Lateral Flow Assay (LFA), *Coccidioides* IDCF immunodiffusion assay (IDCF), and the Clarus *Coccidioides* Ab Enzyme Immunoassay (EIA). Assays were evaluated using 98 dog serum samples: 29 from dogs with coccidioidomycosis, 15 from dogs diagnosed with histoplasmosis, 10 from dogs diagnosed with blastomycosis, and 44 from dogs without a fungal disease. Using specimens from dogs with coccidioidomycosis, the IDCF had an accuracy of 92% (95% confidence interval [95% CI] = 85–96%), the EIA had an accuracy of 91% (95% CI = 83–96%), and the LFA displayed an accuracy of 82% (95% CI = 73–89%). Using Kappa analysis, the agreement between LFA and EIA was 0.59 (95% CI = 0.42–0.75), that between LFA and IDCF was 0.64 (95% CI = 0.48–0.79), and that between EIA and IDCF was 0.79 (95% CI = 0.64–0.90). Most cross-reactions were observed in dogs with histoplasmosis. Compared with EIA and IDCF, the LFA requires substantially less laboratory equipment and infrastructure and rapidly produces results, offering a substantial improvement for the initial screening of coccidioidomycosis in dogs.

## 1. Introduction

Coccidioidomycosis is a fungal disease that is caused by the thermally dimorphic fungus *Coccidioides* spp. The disease is most frequently observed in the Southwestern United States (U.S.), Northern Mexico, and Central and South America [1,2,3,4,5,6]. The endemic range of coccidioidomycosis in North America is expanding and now entails South Central Washington State and Northeastern Utah [1,3,7].

In 2019, over 20,000 human cases of coccidioidomycosis were reported in the U.S. by the Centers for Disease Control and Prevention (CDC); most were reported in Arizona and California, among people older than 60 years [7,8,9]. It is suspected that the true coccidioidomycosis burden is underestimated and that the low rate of laboratory testing contributes to this underestimation [8]. Additionally, rates of coccidioidomycosis among non-humans are unknown [2,4].

Coccidioidomycosis affects a variety of mammalian species, including various domestic animal [10,11,12,13,14,15,16]. As in humans, dogs can suffer from serious disseminated infection [10,11]. An early and accurate diagnosis of coccidioidomycosis and subsequent early initiation of specific antifungal treatment are important in reducing coccidioidomycosis morbidity and mortality. Furthermore, the accurate identification of coccidioidomycosis cases in dogs may also provide a better understanding of the geographical distribution of *Coccidioides* spp. and its epidemiology. Dogs are less likely to travel and be exposed to *Coccidioides* spp. outside their home range compared with humans, thus providing more accurate information.

Conventional culture and histopathology, including special stains, are considered the “standard” for the diagnosis of coccidioidomycosis. However, these assays have several limitations, including the need for biosafety level 3 laboratory infrastructure and highly trained staff. Furthermore, the analytical performance of these assays is variable, and the turnaround time for reporting results can be prolonged, ranging from 7 to 14 days [17].

The detection of anti-*Coccidioides* antibodies (Ab) is traditionally performed using immunodiffusion (ID) and complement fixation (CF). More recently, the enzyme immunoassay (EIA) has become commercially available [18,19]. While the EIA was developed to diagnose coccidioidomycosis in human specimens, it has been successfully modified to identify IgG antibodies against *Coccidioides* spp. in sera from dogs and other mammalian species [20]. EIA shows a good analytical performance and reduces the time to diagnose dogs; a complex laboratory infrastructure and highly trained laboratorians are required to perform EIA [20,21].

Recently, a lateral flow assay (LFA) has been developed to detect total antibody against *Coccidioides* in human serum. This relatively new methodology has multiple advantages, particularly in fungal diagnostics. It is rapid, easy to perform, and requires no refrigeration. Therefore, it can be used in remote locations [22]. The purpose of the current study was to compare three immunodiagnostic assays for the detection of *Coccidioides* antibodies in sera from dogs.

## 2. Materials and Methods

**Dog Sera:** A total of 98 dog sera were tested in this study. Of these, 29 were from dogs diagnosed with coccidioidomycosis. The remaining 69 sera were obtained as non-coccidioidomycosis controls: 23 sera from healthy dogs that resided in areas not known to be endemic for coccidioidomycosis, 10 sera from dogs with blastomycosis, provided by Dr. Gene Scalarone, Idaho State University, and 36 dog sera provided by Dr. Andrew Hanzlicek, Oklahoma State University. These 36 sera included: 15 sera from dogs with histoplasmosis (14 proven and 1 probable) and 21 sera from dogs with non-infectious causes of disease, including various malignancies and metabolic diseases (Figure 1). The sera from the 29 dogs previously diagnosed by serology for coccidioidomycosis using immunodiffusion and a modified EIA test and from the 23 healthy dogs were residual specimens from a previous study [20]. Specimens were stored at −80 °C from 2015 to the time of testing in November 2019.

***Coccidioides* Antibody Immunodiagnostics Assays:** three commercially available immunodiagnostic assays produced by IMMY^®^ (Norman, OK, USA) were evaluated: the sōna *Coccidioides* Ab Lateral Flow Assay (*Coccidioides* Ab LFA), the *Coccidioides* IDCF immunodiffusion assay (*Coccidioides* IDCF ID), and the Clarus *Coccidioides* Ab Enzyme Immunoassay (*Coccidioides* IgG EIA). Assays were performed as follows:

***Coccidioides* Ab LFA**: The assay was performed according to the manufacturer’s instructions. Briefly, dog sera were diluted 1:441 with the manufacturer-provided specimen diluent buffer using flat bottom tubes. The LFA strip was placed in a well containing 100 µL of diluted specimen, and the results were read after 30 min by two independent readers. On the LFA test strip, two lines may be observed, a control and a test line. Upon reading the results, a pink to red color change on the control line must be present to validate results. The lack of a color change on the control line invalidated the test and indicated that the test had to be repeated. The presence of two pink or red lines, at both the control and test lines, was considered a positive result. The presence of a color change at only the control line was considered a negative result. The presence of a gray line at the test line was considered a negative result.

***Coccidioides* IDCF ID****:** The assay was performed according to the manufacturer’s instructions. Briefly, 25 µL of dog sera and the assay’s reference serum were placed into appropriate outer wells of the immunodiffusion plate (Cleargel™ Plates; IMMY, Norman, OK, USA) and pre-incubated at room temperature for 30 minutes. Next, the *Coccidioides* IDCF antigen was added to the center well. The plates were incubated at room temperature in a humidified chamber. After 48 h, the ID plates were read for the presence of a precipitin band located between the wells that received serum and the well that received *Coccidioides* antigen. The presence of a band was recorded and was classified into three categories: band of identity, band of partial-identity, and band of non-identity. As specified by the manufacturer, a band of identity or a band of partial identity were considered positive for *Coccidioides* IDCF antibody. The absence of bands or non-identity band reactions were regarded as a negative test.

***Coccidioides* IgG EIA: The assay** was performed following the modifications described by Chow et al. [20]. Dog sera was diluted 1:25 in specimen diluent buffer provided with the EIA kit. One hundred microliters of diluted sample, CF calibrator cutoff, and positive and negative controls were placed into microwells coated with CF antigen and incubated at room temperature for 30 minutes. After incubation, the plates were washed three times using 1X wash buffer using an EIA plate washer (Model ELx50; Biotek, Winooski, VT, USA). Excess wash buffer was removed from the microwells by striking the plates on a clean paper towel until dry. Peroxidase-conjugated protein A/G (Thermo Fisher Scientific, Rockford, IL, USA) was diluted 1:10,000 in 1X PBS, and 100 µL was added to all wells. After incubation for 30 minutes at room temperature, the plates were washed three times as described above. One hundred microliters of TMB Substrate was added to each well and incubated for 10 minutes at room temperature to allow for color development. Finally, 100 µL of stop solution was dispensed to all wells. The optical density (OD) of each microtiter well was obtained using a dual-wavelength plate reader at 450 nm and 630 nm (Model ELx800; Biotek, Winooski, VT, USA). The calculation of the *Coccidioides* IgG EIA units was obtained by dividing the OD value of serum and control wells by the blanked OD from the calibrator cutoff well. Based on Chow et al. [20], samples with an EIA unit value ≥1.33 were considered positive, and samples below 1.33 EIA units were considered negative.

**Statistical analysis***:* The analytical performances and their respective 95% confidence intervals (95% CI) were calculated using 2 × 2 tables. Kappa values and their respective 95% CI were calculated. The *Coccidioides* Ab immunodiagnostic assay’s agreement was evaluated using the Kappa index, and this index was interpreted as follow: 0.0 to 0.2, no agreement; 0.21 to 0.39, minimal agreement; 0.40 to 0.59, weak agreement; 0.60 to 0.79, moderate agreement; 0.80 to 0.89, strong agreement; and ≥0.90, perfect agreement [23,24]. Analyses were conducted using STATA 11.

## 3. Results

***Coccidioides* Ab LFA***:* Of the 29 sera from dogs diagnosed with coccidioidomycosis, 25 were positive by the *Coccidioides* Ab LFA (sensitivity 86%, 95% CI 68–96%). Of the 69 non-coccidioidomycosis control sera, 14 were positive by LFA (specificity of 80%, 95% CI 68–88%). Of these 14 sera, one (10%) was from a dog with blastomycosis, eight (53%) from dogs with histoplasmosis, and five (24%) from dogs that had a non-infectious cause of disease. No positive results were observed among sera from the 23 healthy dogs residing in regions not known to be endemic for coccidioidomycosis. The assay accuracy was 82% (95% CI 73–89%) (Figure 2 and Figure 3, and Table 1).

***Coccidioides* IDCF ID:** Of the 29 sera from dogs diagnosed with coccidioidomycosis, 25 had positive bands of identity (sensitivity 86%, 95% CI 68–96%). Of the 69 non-coccidioidomycosis control sera, 61 displayed negative results (specificity of 94%; 95% CI 86–98%). Of the remaining eight sera, four displayed bands of partial-identity, and four displayed bands of non-identity (considered negative results). Three of the sera displaying bands of partial identity, considered positive results, were observed in specimens from dogs diagnosed with histoplasmosis, and one was observed in the specimen of a dog that had a non-infectious cause of disease. Of the four sera that displayed bands of non-identity, three were from dogs with histoplasmosis and one from a dog that had a non-infectious cause of disease. The assay accuracy was 92% (95% CI 85–96%) (Figure 2 and Figure 3, and Table 1).

***Coccidioides* IgG EIA:** Of the 29 sera from dogs with coccidioidomycosis, 27 were positive by *Coccidioides* IgG EIA (sensitivity of 93%, 95% CI 77–99%). Of the 69 specimens from dogs not infected with *Coccidioides*, 62 displayed negative results (90% specificity; 95% CI 80–69%). Seven false-positive EIA results were observed: one from a healthy dog residing in an area not known to be endemic for coccidioidomycosis, three from dogs with histoplasmosis, and three from dogs that had a non-infectious cause of disease. The assay accuracy was 91% (95% CI 83–96%) (Figure 2 and Figure 3, and Table 1).

**Concordance analysis:** The concordance between LFA readers was 0.940 (kappa index 95% CI 0.870–1.000). Reader discrepancy was observed in one sample from the 29 dogs diagnosed with coccidioidomycosis, and in two samples from the 69 dogs without coccidioidomycosis.

**Concordance analysis in dogs with coccidioidomycosis:** When comparing the different assays for the detection of *Coccidioides* Ab in 29 dogs with coccidioidomycosis, 22 sera were positive by all three *Coccidioides* Ab detection assays (LFA, EIA, IDCF). Of the remaining seven sera that were not positive by all three assays, three were positive by both EIA and IDCF, two were positive by both LFA and EIA, one by LFA alone, and one serum was negative by all three assays for Ab detection. These results represented four false negatives by LFA, three false negatives by IDCF assay, and two false negatives by EIA (Figure 3).

**Concordance analysis in dogs without coccidioidomycosis:** Of the 69 dogs without coccidioidomycosis, the sera of 51 dogs displayed negative results by all three assays, and sera from 18 tested positive. Of these eighteen false-positive results, nine were LFA only, four were EIA only, and no false-positive results were observed by IDCF only. Of the remaining five sera that displayed false-positive results, two were falsely positive by all three antibody assays, and three were falsely positive by two of the antibody assays (two by LFA and IDCF, and one by LFA and EIA) (Figure 3).

**Concordance analysis showed the following Kappa index values:** *Coccidioides* Ab LFA/EIA: Kappa = 0.586 (95% CI 0.421–0.751), considered a weak agreement; *Coccidioides* Ab LFA/IDCF: Kappa = 0.644 (95% CI 0.482–0.792), considered a moderate agreement; and *Coccidioides* Ab EIA/IDCF ID: 0.790 (95% CI 0.637–0.904), considered a moderate agreement (Figure 3).

## 4. Discussion

Traditionally, serodiagnosis of coccidioidomycosis has relied on the standard assays of CF and ID. These assays require highly trained laboratorians and two to three days to obtain results [17]. Newer, more rapid assays have been developed in recent years and have shown promise in reducing the time to obtain results. This study was undertaken to determine how these methods perform for the detection of anti-*Coccidioides* antibodies, using sera from dogs with and without coccidioidomycosis. The prompt and accurate identification of canine anti-*Coccidioides* antibodies could decrease the time to diagnosis in symptomatic dogs and may reduce complications and mortality by decreasing the time to initiating antifungal treatment. Additionally, an accurate diagnosis in dogs could help to better define the epidemiology of coccidioidomycosis and geographical distribution of *Coccidioides* spp., since dogs are less likely leave their home range, limiting their location of exposure. It is also important to remember that these immunodiagnostic methods are tools used for case definition. The case definition of coccidioidomycosis also requires a correlation of these laboratory findings with the clinical presentation and epidemiology of the canine patient.

Twenty-two of the twenty-nine dogs with coccidioidomycosis were reactive with all three assays, resulting in a positive concordance of 75%. Of the three assays, IgG EIA was determined to be the most sensitive (93%). LFA and ID assays displayed an equal sensitivity of 86%. One dog with coccidioidomycosis was negative by all assays, possibly due to the limit of detection of the assays or a lack of Ab in the animal tested. The sensitivities reported in this study for the EIA and the IDCF were similar to those reported by Chow et al. [20], where the sensitivity was 95% and specificity was 96%. As well as providing an excellent sensitivity, the IgG EIA also delivered a turn-around time of three hours.

The assay that displayed the best specificity was the IDCF with a 94% specificity, followed by the IgG EIA and LFA with a specificity of 90% and 80%, respectively. Of the three groups of dogs without coccidioidomycosis, the least amount of false-positive reactivity was observed in the healthy dogs and the dogs with blastomycosis, with only one false-positive result from each. The greatest number of false-positive results were in dogs diagnosed with histoplasmosis, across all three assays. This is not totally unexpected, as cross-reactivity between *Histoplasma* and *Coccidioides* has been demonstrated in other reports [18,19]. This also agrees with the manufacturer’s package insert for the *Coccidioides* EIA, which reported a false positivity rate of 80% in specimens from human patients with histoplasmosis [22]. When evaluating all other non-histoplasmosis dogs, the specificity for EIA, IDCF, and LFA was 92%, 98%, and 88%, respectively. Compared with data reported by Chow et al., EIA and IDCF showed a similar specificity in canine specimens [20].

Of all the dog sera that had reactive bands by IDCF, four sera displayed bands of partial-identity, three from dogs with histoplasmosis and one from a dog that had a non-infectious cause of disease. As instructed by the manufacturer, a band of partial identity should be interpreted as a positive result. However, all partial-identity bands observed in this study were false-positive results, mostly in dogs with histoplasmosis. Furthermore, in this study, no bands of partial identity were observed in any of the dogs with coccidioidomycosis. Bands of non-identity, considered a negative result, were not observed in any of the dogs with coccidioidomycosis. Based on this result, the presence of partial-identity bands should be interpreted carefully, and results should be correlated with clinical findings and epidemiology.

A few studies have described the evaluation of serological assays in dogs for the diagnosis of coccidioidomycosis. Reports include the evaluation of EIA for the detection of IgG and more recently the evaluation of a novel LFA detection method [20,21,25,26]. The sensitivity of EIA has been shown to be greater than 90% [20,21]. A major limitation is the lack of a commercial assay to provide accurate and reproducible results. Additionally, a sufficient laboratory capacity with well-trained lab personnel is required [20,21]. Recent studies have compared this LFA against immunodiffusion [25,26]; as we observed in this study, the LFA presented a moderate to good agreement with the immunodiffusion test [25,26]. This is the first study to evaluate and compare three different Ab detection assays.

Based on the Kappa index values, we observed that LFA presented a moderate agreement with the ID and a weak agreement with the EIA. Likewise, a moderate agreement was observed between the ID and the EIA. These agreements were mostly affected by the assays’ specificities. All three assays displayed a high negative predictive value.

The limitations of this study included: the small sample size and lack of access to complete information from clinical records on the associated complications and mortality of the dogs tested. This study only evaluated products from IMMY and did not include assays for the detection of *Coccidioides* IgM antibodies by ID or EIA that would provide a better comparison with the LFA, which is designed to detect both IgG and IgM antibodies for *Coccidioides*. Lastly, further evaluations in other animal species and using other commercially available products are needed. Further investigations using prospective cohort studies of dogs with a good clinical and epidemiological characterization are needed.

## 5. Conclusions

This study compared three commercial *Coccidioides* immunodiagnostics assays for the diagnosis of coccidioidomycosis in dogs, including an IDCF assay, a modified EIA for *Coccidioides* IgG, and a point of care LFA test for anti-*Coccidioides* Ab. The analysis of the three assays demonstrated that each test displayed a strong analytical performance, with accuracies higher than 80%. While the EIA provided the best sensitivity, it required the modification of a commercially available assay, along with a need for specialized equipment and well-trained laboratory professionals. Conversely, the IDCF was the most specific, but the results were not available for 48–72 h. In comparison with EIA and IDCF, the LFA was performed with minimal laboratory equipment and laboratory infrastructure and provided results in less than one hour, faster than other immunodiagnostic assays. Even though the accuracy of the assay may be affected by the specificity, it is a promising assay that may be incorporated into lower-complexity laboratories that may not have the capability to perform IDCF or EIA.

## Figures and Tables

**Figure 1 jof-08-00728-f001:**
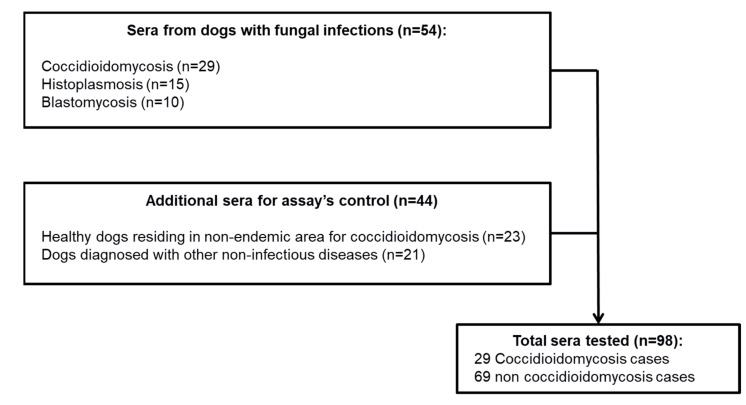
Flow chart of serum samples tested for the validation of the immunodiagnostic assays for the rapid diagnosis of coccidioidomycosis in dogs.

**Figure 2 jof-08-00728-f002:**
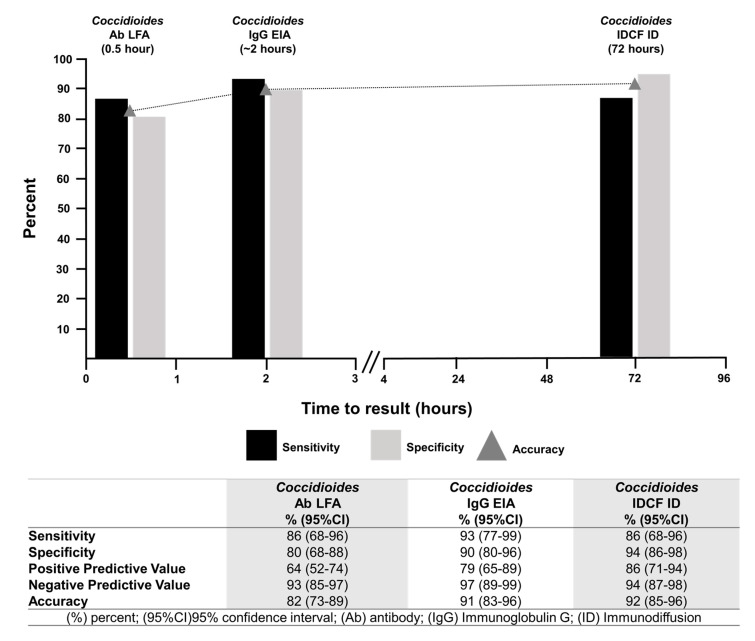
Comparison of the sensitivity, specificity, analytical performance, and time to results with the IMMY^®^ sōna *Coccidioides* total antibodies (Ab) lateral flow assay (LFA), modified IMMY^®^ *Coccidioides* Ab Enzyme Immunoassay (EIA), and the IMMY^®^ IDCF immunodiffusion for the detection of anti-*Coccidioides* IgG Ab in dog serum.

**Figure 3 jof-08-00728-f003:**
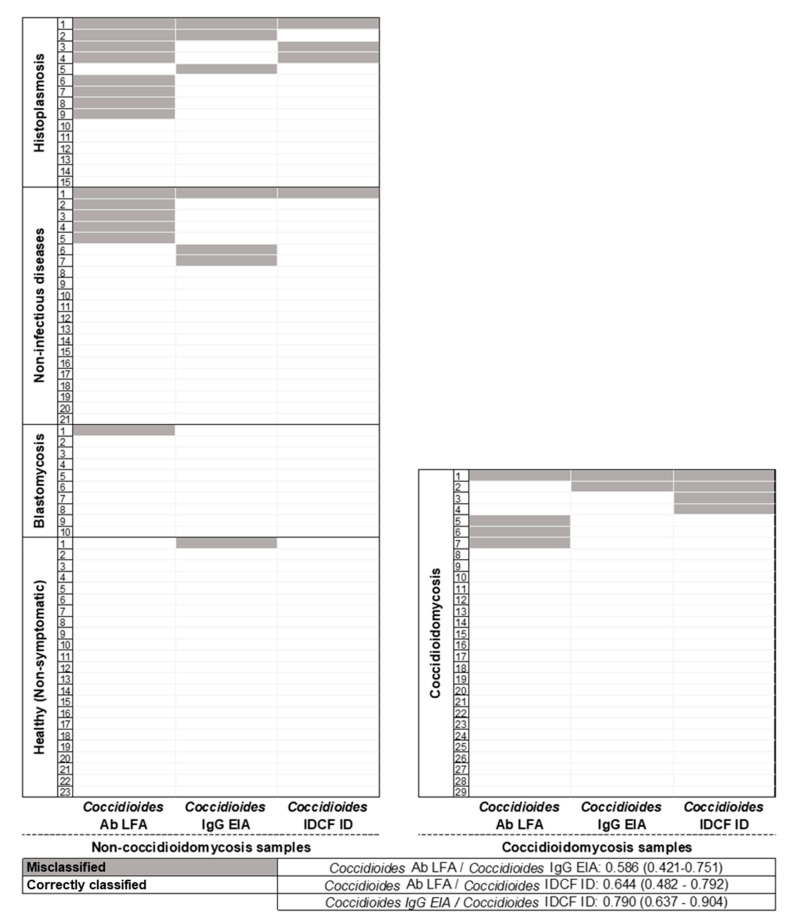
Concordance analysis of assays for the detection of *Coccidioides* Ab: Analysis of 98 dog sera tested using the IMMY^®^ sōna *Coccidioides* lateral flow assay for the detection of total anti-*Coccidioides* Ab (*Coccidioides* Ab LFA), the modified IMMY^®^ *Coccidioides* Enzyme Immunoassay for the detection of anti-*Coccidioides* IgG Ab (*Coccidioides* IgG EIA), and the IMMY^®^ IDCF immunodiffusion for the detection of anti-*Coccidioides* IgG Ab (*Coccidioides* IDCF ID).

**Table 1 jof-08-00728-t001:** Comparison of sensitivity and specificity of *Coccidioides* antibody detection assays.

Assay Sensitivity	Total n	EIA-Positive n (%)	ID-Positive n (%)	LFA-Positive n (%)
*Coccidioides*-infected dogs	29	27 (93)	25 (86)	25 (86)
**Assay Specificity**	**Total** **n**	**EIA-Positive** **n (%)**	**ID-Positive** **n (%)**	**LFA-Positive** **n (%)**
Healthy dogs	23	1 (96)	0 (100)	0 (100)
*Histoplasma*-infected dogs	15	3 (80)	3 (80)	8 (47)
*Blastomyces*-infected dogs	10	0 (100)	0 (100)	1 (90)
Non-infectious disease dogs	21	3 (86)	1 (95)	5 (76)
**Total non-*Coccidioides* infected dogs**	**69**	**7 (89)**	**4 (94)**	**14 (80)**

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
