# Peer review of "Comparison of Immunodiagnostic Assays for the Rapid Diagnosis of Coccidioidomycosis in Dogs"

_jof, 2022, doi:10.3390/jof8070728_

Round 1

Reviewer 1 Report

This manuscript compare three commercial serological methods for coccidioidomycosis diagnosis in dogs. The assays showed strong analytical performance and good accuracies.

There are some specific comments and questions that I have listed below.

Materials and Methods

Page 2 Lines 80-82

I suggest that the authors indicate  the serological test used for the 29 dogs diagnosed with coccidioidomycosis in the previous work.

When the 29 positive samples were defrosted, were they retested using  original serology test ?

How long the sera were stored at -80 °C?

Page 3 Lines 113-116

According to the manufacturer, I would add "absence of bands or non-identity reactions are regarded as a negative test"

Results

Page 4 Lines 157-165

Non-identity band is a negative ID result, hence, I suggest rewriting the sentence and simplified.

"Only 4/69 control sera without coccidioidomycosis were positive by ID test, these displayed partial-identity (describe)"

 Page 5 Lines 189-190

“one serum was negative by all three assays for Ab detection”

Could this sample be denatured due to storage time in sera collection? Was this possibility ruled out?

Discussion

Page 7 Lines 253-255

In agreement with EORTC/MSGERC-2019 anything serological positive result (identity and partial- identity) should be correlated with evidence of geographical/occupational exposure to the fungus and clinical findings.

Line 257

You have to write "coccidioidomycosis" instead of "histoplasmosis"

Lines 263-264

Check the paragraph (after the period).

Reference

Check reference 23 format.  Is Available in the web?

Author Response

Reviewer 1.

This manuscript compare three commercial serological methods for coccidioidomycosis diagnosis in dogs. The assays showed strong analytical performance and good accuracies.

There are some specific comments and questions that I have listed below.

Materials and Methods

Page 2 Lines 80-82

I suggest that the authors indicate  the serological test used for the 29 dogs diagnosed with coccidioidomycosis in the previous work. 

R. Thank you, we added the description of test used.

When the 29 positive samples were defrosted, were they retested using original serology test?

R. Yes, we repeated the original testing.

How long the sera were stored at -80 °C?

R. From 2015 to 2018

According to the manufacturer, I would add "absence of bands or non-identity reactions are regarded as a negative test"

R. Thank you, we added the statement suggested

Results

Page 4 Lines 157-165. Non-identity band is a negative ID result, hence, I suggest rewriting the sentence and simplified.

R. Thank you for your comment. This is address on the next question.

"Only 4/69 control sera without coccidioidomycosis were positive by ID test, these displayed partial-identity (describe)"

R. We added results interpretation of non-identity and partial identity bands.

Page 5 Lines 189-190. “one serum was negative by all three assays for Ab detection”

Could this sample be denatured due to storage time in sera collection? Was this possibility ruled out? 

R. No, the specimen was also negative in the previous study. Looks like this dog did not had antibodies at the moment of specimen collection.

Discussion

Page 7 Lines 253-255. In agreement with EORTC/MSGERC-2019 anything serological positive result (identity and partial- identity) should be correlated with evidence of geographical/occupational exposure to the fungus and clinical findings.

R. Thank you for your comment

Line 257. You have to write "coccidioidomycosis" instead of "histoplasmosis"

R. Thank you, we corrected it.

Lines 263-264. Check the paragraph (after the period).

Reference

Check reference 23 format.  Is Available in the web?

R. No, it is not available in the web.

Reviewer 2 Report

This is a well-written manuscript showing a comparison of immunodiagnosis assays for fast anti-Coccidioides antibody detection in dogs.  This study compared three commercial Coccidioides immunodiagnostics assays for the diagnosis of coccidioidomycosis in dogs: a traditional IDCF assay, a modified EIA for Coccidioides IgG, and a point of care test Coccidioides Ab LFA. The study is novel and brings important issues regarding diagnosis of coccidioicomycosis in dogs. First, accurate diagnosis could provide substantial improve on the quality of canine life, as well as, to advance mapping of endemic regions of coccidioidomycosis, as stated by the authors.  However, the main issue was the demonstration of high accuracy in the commercial assays tested, where EIA provide the best sensitivity, and IDCF best specificity, but both or require specialized equipment or is time-consuming. Therefore, LFA is recommended for initial screening of coccidioidomycosis in dogs since requires less laboratory infrastructure and rapidly generates results. Overall, this study´s approach is generally sound. The paper is well structured and the objectives are clear. The manuscript represents an interesting and valuable record and therefore is suitable for publication.

Author Response

Reviewer 2

This is a well-written manuscript showing a comparison of immunodiagnosis assays for fast anti-Coccidioides antibody detection in dogs.  This study compared three commercial Coccidioides immunodiagnostics assays for the diagnosis of coccidioidomycosis in dogs: a traditional IDCF assay, a modified EIA for Coccidioides IgG, and a point of care test Coccidioides Ab LFA. The study is novel and brings important issues regarding diagnosis of coccidioicomycosis in dogs. First, accurate diagnosis could provide substantial improve on the quality of canine life, as well as, to advance mapping of endemic regions of coccidioidomycosis, as stated by the authors.  However, the main issue was the demonstration of high accuracy in the commercial assays tested, where EIA provide the best sensitivity, and IDCF best specificity, but both or require specialized equipment or is time-consuming. Therefore, LFA is recommended for initial screening of coccidioidomycosis in dogs since requires less laboratory infrastructure and rapidly generates results. Overall, this study´s approach is generally sound. The paper is well structured and the objectives are clear. The manuscript represents an interesting and valuable record and therefore is suitable for publication.

R. Thank you.

Reviewer 3 Report

This is a compact, generally well-written manuscript comparing various serological diagnostic tests in dogs. The methodology is very straight-forward and they strive to address the limitations of tests and cross-reactivity while demonstrating tests' agreement with each other and comparing sensitivity and specificity. The tests are all produced by one company.

1) There are numerous typographical errors, dropped words, inconsistent use of italics/not italics for Coccidioides, and some bad sentences throughout this document.  I strongly urge the authors to go through it very carefully.  Some sentences do not make sense.  I will try to point out the most glaring.  All of it is minor and easily fixed.

2) Though they test all of the assays against serum from dogs with related geographically limited diseases, they don't address in either the introduction or the discussion that this is relevant as it relates to travel history and location of acquisition of dogs.  When discussing the cross-reactivity of tests as relates to their specificity, this seems important if the purpose is a diagnostic test for clinical disease and not an epidemiological tool within the endemic area.

Lines 63-64 - this sentence does not make sense.  The summation of this paragraph is that all the tests require laboratory infrastructure and skilled people. 

LIne 69 - in a veterinary setting, it can be used in any practice in a bedside manner. There are two published references that have described this potential use of the LFA in dogs (your references 25 and 26).

Line 74 - I think this requires additional information on how the dogs were diagnosed with coccidioidomycosis originally.  Since you are testing serology tests, were the only diagnostics done to diagnose coccidioidomycosis in the 29 dogs serology tests?  What serology tests?

Line 77 - for my own edification, I am curious if they have blastomycosis in Idaho?  

Lines 113-116 - there are some errors in these sentences regarding the use of was/were. Please correct them.

Table 1 is very nice!

Line 161 specimens should be plural

LIne 195, comma not a semicolon 

Line 19 needs an 'and' after the comma as they are not parallel structures.

216 and 217 - what are the newer methods that don't required skilled laboratorians? Maybe be specific and provide references.  Also, did these tests provide accurate results?  What about, for example, a dog in Oklahoma with a positive cocci LFA but it came from a shelter in Phoenix or MIdland? What diagnostics to do next?  Though treatment of cocci and histoplasma at least start out the same, time of treatment, recurrence rate, and second line treatment for failure may differ for the two diseases.  It would be helpful to differentiate clinically.

Line 229 - see Gunstra, et.al., on seronegativity in histologically diagnosed dogs with cocci

I think the discussion should address why the dogs with non-infectious other diseases might have a high rate of false positives compared to healthy dogs. It seemed to occur for 2 of the three tests evaluated.

This reviewer also thinks it is more of a limitation of this manuscript than the authors do that there is no clinical or other information regarding diagnosis of the 29 dogs with coccidioidomycosis.  Reference 20 states these were residual samples from Mira Vista Diagnostics, so they were "tested" with one or more commercial assays from a different laboratory, but there is no information on confirmation of diagnosis, and whether they were diagnosed by Ag or Ab detection or both.   Consider expanding on this limitation a bit since you tested specimens with no clinical information and for which there are limitations of a similar nature for the assays that may have been used to diagnose coccidioidomycosis in them.

Line 264 - word is missing.

Overall, this is a good first step toward helping to improve serological diagnosis of coccidioidomycosis in dogs.  In particular, the information about possibility of cross-reactivity and false positives in "sick" dogs without an established infectious diagnosis are interesting. 

Author Response

Reviewer 3

This is a compact, generally well-written manuscript comparing various serological diagnostic tests in dogs. The methodology is very straight-forward and they strive to address the limitations of tests and cross-reactivity while demonstrating tests' agreement with each other and comparing sensitivity and specificity. The tests are all produced by one company.

There are numerous typographical errors, dropped words, inconsistent use of italics/not italics for Coccidioides, and some bad sentences throughout this document.  I strongly urge the authors to go through it very carefully.  Some sentences do not make sense.  I will try to point out the most glaring.  All of it is minor and easily fixed.

R. Thank you, we reviewed and fixed the typographical errors.

Though they test all of the assays against serum from dogs with related geographically limited diseases, they don't address in either the introduction or the discussion that this is relevant as it relates to travel history and location of acquisition of dogs.  When discussing the cross-reactivity of tests as relates to their specificity, this seems important if the purpose is a diagnostic test for clinical disease and not an epidemiological tool within the endemic area.

R. Agree, looks like this a useful too for lab diagnosis. Unfortunately, we had limited access to dogs record.

Lines 63-64 - this sentence does not make sense.  The summation of this paragraph is that all the tests require laboratory infrastructure and skilled people.

R. Thank you for your comment. We reviewed the manuscript.

LIne 69 - in a veterinary setting, it can be used in any practice in a bedside manner. There are two published references that have described this potential use of the LFA in dogs (your references 25 and 26).

R. Yes, it would be a useful tool for screening in veterinary setting

Line 74 - I think this requires additional information on how the dogs were diagnosed with coccidioidomycosis originally.  Since you are testing serology tests, were the only diagnostics done to diagnose coccidioidomycosis in the 29 dogs serology tests?  What serology tests?

R. Unfortunately we did not had access to dogs clinical record. Previous diagnose was done by immunodiffusion and EIA. We clarified it in the manuscript.

Line 77 - for my own edification, I am curious if they have blastomycosis in Idaho? 

R. No, it is not considered endemic region. Probably Dr Dr. Gene Scalarone at Idaho State University got these specimens as port of his research projects.

Lines 113-116 - there are some errors in these sentences regarding the use of was/were. Please correct them.

R. Thank you, we reviewed it.

Table 1 is very nice!

R. Thank you.

Line 161 specimens should be plural

R. Thank you, we corrected it.

Line 195, comma not a semicolon

R. Thank you, we corrected it.

Line 19 needs an 'and' after the comma as they are not parallel structures.

R. Something is missing on this question.

216 and 217 - what are the newer methods that don't required skilled laboratorians? Maybe be specific and provide references.  Also, did these tests provide accurate results?  What about, for example, a dog in Oklahoma with a positive cocci LFA but it came from a shelter in Phoenix or MIdland? What diagnostics to do next?  Though treatment of cocci and histoplasma at least start out the same, time of treatment, recurrence rate, and second line treatment for failure may differ for the two diseases.  It would be helpful to differentiate clinically.

R. Thank you for your comment. We added and statement remembering the role of immunodiagnostics on coccidioidomycosis case definition.

Line 229 - see Gunstra, et.al., on seronegativity in histologically diagnosed dogs with cocci

I think the discussion should address why the dogs with non-infectious other diseases might have a high rate of false positives compared to healthy dogs. It seemed to occur for 2 of the three tests evaluated.

This reviewer also thinks it is more of a limitation of this manuscript than the authors do that there is no clinical or other information regarding diagnosis of the 29 dogs with coccidioidomycosis.  Reference 20 states these were residual samples from Mira Vista Diagnostics, so they were "tested" with one or more commercial assays from a different laboratory, but there is no information on confirmation of diagnosis, and whether they were diagnosed by Ag or Ab detection or both.   Consider expanding on this limitation a bit since you tested specimens with no clinical information and for which there are limitations of a similar nature for the assays that may have been used to diagnose coccidioidomycosis in them.

R. We added an additional statement on study limitation, and we also complemented it with a suggestion for future research.

Line 264 - word is missing.

R. Thank you, we corrected the statement.

Overall, this is a good first step toward helping to improve serological diagnosis of coccidioidomycosis in dogs.  In particular, the information about possibility of cross-reactivity and false positives in "sick" dogs without an established infectious diagnosis are interesting.

R. Thank you.

Reviewer 4 Report

I

In the article “Comparison of Immunodiagnostic Assays for the Rapid Diagnosis of Coccidioidomycosis in Dogs,” the authors compare three immunodiagnostic assays for the detection of anti-Coccidioides antibodies in dogs. The article is interesting and of great application for diagnosing coccidioidomycosis, where the three tests presented a solid analytical performance, with accuracies more significant than 80%.

Likewise, it is a well-structured manuscript; the content is consistent with the title and represents an essential contribution to the diagnosis of coccidioidomycosis. I only recommend reviewing the article as it has some typos.

Author Response

Reviewer 4

In the article “Comparison of Immunodiagnostic Assays for the Rapid Diagnosis of Coccidioidomycosis in Dogs,” the authors compare three immunodiagnostic assays for the detection of anti-Coccidioides antibodies in dogs. The article is interesting and of great application for diagnosing coccidioidomycosis, where the three tests presented a solid analytical performance, with accuracies more significant than 80%.

Likewise, it is a well-structured manuscript; the content is consistent with the title and represents an essential contribution to the diagnosis of coccidioidomycosis. I only recommend reviewing the article as it has some typos.

R. Thank you.